# Modeling Cardiotoxicity in Pediatric Oncology Patients Using Patient-Specific iPSC-Derived Cardiomyocytes Reveals Downregulation of Cardioprotective microRNAs

**DOI:** 10.3390/antiox12071378

**Published:** 2023-07-03

**Authors:** Ignacio Reinal, Imelda Ontoria-Oviedo, Marta Selva, Marilù Casini, Esteban Peiró-Molina, Carlos Fambuena-Santos, Andreu M. Climent, Julia Balaguer, Adela Cañete, Jaume Mora, Ángel Raya, Pilar Sepúlveda

**Affiliations:** 1Regenerative Medicine and Heart Transplantation Unit, Health Research Institute Hospital la Fe, 46026 Valencia, Spain; igreifer@upvnet.upv.es (I.R.);; 2Hospital Universitari i Politècnic La Fe, 46026 Valencia, Spaincanyete_ade@gva.es (A.C.); 3ITACA Institute, Universitat Politècnica de València, 46026 Valencia, Spainacliment@itaca.upv.es (A.M.C.); 4Transtational Research in Cancer Unit-Pediatric Oncology, Health Research Institute Hospital La Fe, 46026 Valencia, Spain; 5Department of Pediatrics, University of Valencia, 46010 Valencia, Spain; 6Oncology Service, Hospital Sant Joan de Déu, 08950 Esplugues de Llobregat, Spain; 7Regenerative Medicine Program, Bellvitge Biomedical Research Institute (IDIBELL), L’Hospitalet de Llobregat, 08908 Barcelona, Spain; araya@idibell.cat; 8Program for Clinical Translation of Regenerative Medicine in Catalonia—P-[CMRC], L’Hospitalet de Llobregat, 08908 Barcelona, Spain; 9Centro de Investigación Biomédica en Red Bioingeniería, Biomateriales y Nanomedicina (CIBER-BBN), Carlos III Institute of Health, 28029 Madrid, Spain; 10Institució Catalana de Recerca i Estudis Avançats (ICREA), 08010 Barcelona, Spain; 11Centro de Investigación Biomédica en Red Enfermedades Cardiovasculares (CIBERCV), Carlos III Institute of Health, 28029 Madrid, Spain; 12Department of Pathology, University of Valencia, 46010 Valencia, Spain

**Keywords:** iPSC-derived cardiomyocytes, cardiotoxicity, pediatric patients, oxidative stress, doxorubicin, microRNA

## Abstract

Anthracyclines are widely used in the treatment of many solid cancers, but their efficacy is limited by cardiotoxicity. As the number of pediatric cancer survivors continues to rise, there has been a concomitant increase in people living with anthracycline-induced cardiotoxicity. Accordingly, there is an ongoing need for new models to better understand the pathophysiological mechanisms of anthracycline-induced cardiac damage. Here we generated induced pluripotent stem cells (iPSCs) from two pediatric oncology patients with acute cardiotoxicity induced by anthracyclines and differentiated them to ventricular cardiomyocytes (hiPSC-CMs). Comparative analysis of these cells (CTX hiPSC-CMs) and control hiPSC-CMs revealed that the former were significantly more sensitive to cell injury and death from the anthracycline doxorubicin (DOX), as measured by viability analysis, cleaved caspase 3 expression, oxidative stress, genomic and mitochondrial damage and sarcomeric disorganization. The expression of several mRNAs involved in structural integrity and inflammatory response were also differentially affected by DOX. Functionally, optical mapping analysis revealed higher arrythmia complexity after DOX treatment in CTX iPSC-CMs. Finally, using a panel of previously identified microRNAs associated with cardioprotection, we identified lower levels of miR-22-3p, miR-30b-5p, miR-90b-3p and miR-4732-3p in CTX iPSC-CMs under basal conditions. Our study provides valuable phenotype information for cellular models of cardiotoxicity and highlights the significance of using patient-derived cardiomyocytes for studying the associated pathogenic mechanisms.

## 1. Introduction

Early detection of pediatric cancer and improvements in oncology therapies have significantly increased the number of long-term survivors [1,2,3,4]. Anthracyclines are widely-used chemotherapeutic agents effective against a broad spectrum range of malignancies, and are included in >50% of all pediatric chemotherapy regimens [4]; however, their use is hampered by cardiotoxicity. This has resulted in a significant proportion of childhood survivors facing premature cardiac disease, which is a major cause of early morbidity and mortality [5]. It is crucial to consider the impact of age on the development and progression of cardiotoxicity, as variations in disease progression, response to treatment and susceptibility to cardiotoxic effects have been observed between children and adults [6]. Pediatric cardiotoxicity can be acute and occur early during treatment and includes cardiovascular complications such as arrhythmias, transient or progressive left ventricular systolic dysfunction, myocarditis and ischemic disease [1]. Acute cardiotoxicity typically occurs during or shortly after administration of anthracycline treatment. The most common manifestations are transient electrocardiogram (ECG) changes, arrhythmias and acute myocardial injury. Patients may experience symptoms such as chest pain, shortness of breath, palpitations or edema. Delayed cardiotoxicity can occur months to years after completion of anthracycline therapy, with a cumulative dose-dependent effect. It is often characterized by progressive and irreversible cardiac damage, and commonly observed manifestations include left-ventricular systolic dysfunction, reduction in left-ventricular ejection fraction (LVEF) and the development of heart failure. Acute cardiotoxicity is encountered at a much higher frequency in adult patients than in children, most likely owing to the increased frequency of prior cardiac conditions at the time of antineoplastic treatment [3]. Chronic cardiotoxicity in long-term childhood cancer survivors (up to 30 years after cancer treatment) presents with symptomatic cardiac dysfunction in up to 16% of anthracycline-exposed survivors [7].

Although there appears to be a dose–response effect for anthracycline-related cardiotoxicity, some studies have suggested that there is no safe dose of these agents [2,8]. While the mechanism through which anthracyclines induce cardiotoxicity is not fully understood, it typically results in ventricular cardiomyocyte damage. For instance, anthracyclines can induce oxidative stress in cardiomyocytes [9,10], stimulate DNA damage by binding DNA topoisomerase IIβ (Top2b) [11], modify autophagic flux [12,13] or trigger mitochondrial damage [14,15]. It has recently been reported that alterations in the expression of plasma microRNAs in children receiving anthracyclines correlate with markers of cardiac damage [16]. Also, genetic variations in at least 75 genes are associated with anthracycline-related cardiotoxicity [17,18].

When comparing the mechanisms of cardiotoxicity between adult and young cancer patients, several differences arise. Young cancer patients may exhibit higher vulnerability to anthracycline-induced cardiotoxicity due to their ongoing cardiac development and maturation processes. Additionally, if they are diagnosed at an early age, they may experience longer exposure to potential cardiotoxic agents over their lifetime. Furthermore, the genetic factors contributing to cardiotoxicity susceptibility could vary between adult and young patients, reflecting distinct genetic profiles and susceptibility genes [19]. Consequently, there is a need to develop customized cellular models for studying cardiotoxicity in pediatric patients, which will contribute to the advancement of prevention and management strategies in this patient population.

The use of induced pluripotent stem cells (iPSCs) offers a powerful model for studying disease mechanisms associated with specific cell types [20,21], allowing for the identification of key molecular circuits and potential therapeutic targets [22]. In this context, investigating the mechanisms of anthracycline-induced cardiotoxicity can be accomplished using human iPSC-derived ventricular cardiomyocytes (hiPSC-CMs) of patients who develop cardiotoxicity after antineoplastic treatment. Indeed, a groundbreaking study from Burridge et al. showed that hiPSC-CMs derived from patients with breast cancer experiencing anthracycline-induced cardiotoxicity recapitulate anthracycline susceptibility in vitro [23]. Other similar studies have used hiPSC-CMs to examine the underlying cellular mechanisms of trastuzumab-induced cardiotoxicity [24] and to study the cardioprotective role of the solute carrier gene *SLC28A3* in childhood cancer survivors who do not experience cardiotoxicity [25].

In the present study, we generated hiPSCs from two pediatric oncology patients who developed acute cardiotoxicity (CTX hiPSCs) when exposed to anthracyclines. hiPSCs were differentiated into cardiomyocytes to test their susceptibility to doxorubicin (DOX), the most extensively used anthracycline in the treatment of pediatric cancer [3], and were compared them against cardiomyocytes derived from two “control” (CTRL) hiPSC lines. We observed that DOX susceptibility was greater in CTX hiPSC-CMs than in CTRL hiPSC-CMs. Moreover, CTX hiPSC-CMs showed reduced cell viability and increased apoptosis, and had higher levels of reactive oxygen species (ROS) than CTRL hiPSC-CMs. Notably, the levels of sarcomere disorganization, genotoxic damage, mitochondrial damage and arrythmia complexity were also greater in CTX hiPSC-CMs than in CTRL hiPSC-CMs when exposed to DOX. Finally, we observed differences in the expression of several key mRNAs and miRNAs that could explain, at least partly, the evident differences in DOX-induced cardiotoxicity. Specifically, CTX hiPSC-CMs showed reduced levels of miR-4732-3p, which we have previously identified as dysregulated in serum from patients with breast cancer and cardiotoxicity. We suggest that the lower levels of miR-4732-3p in CTX hiPSC-CMs are a potential risk factor for cardiotoxicity, and that this cellular model can be utilized to better understand the mechanism of anthracycline-induced cardiotoxicity and to develop cardioprotective therapies.

## 2. Methods

### 2.1. Patients and Ethical Statements

We recruited two pediatric oncology patients with acute anthracycline-related cardiotoxicity from the Hospital San Joan de Déu (Barcelona, Spain) and Hospital Universitari i Politècnic La Fe (Valencia, Spain). For comparative analyses, we used control cell lines from subjects obtained from the Spanish National Cell Line Bank. Demographic data of the subjects are shown in Table 1. Cardiotoxicity was defined as a decrease in left-ventricular ejection fraction of more than 10%, reaching <50% (reference value for cardiac echocardiography) [26]. The study was approved by the local ethics committee of the Institute of Heath Research La Fe (Cardiocare project, reference 6/2018) and the Comisión de Garantías para la Donación y Utilización de Células y Tejidos Humanos from the Instituto de Salud Carlos III (ref. 470 378 1). The study was conducted in accordance with the Declaration of Helsinki. Parents/guardians of the participants provided written informed consent.

### 2.2. Human iPSC Generation and Maintenance

CTX iPSCs were derived from peripheral blood mononuclear cells. Reprogramming was performed at the Barcelona Stem Cell Bank, Regenerative Medicine Program, Bellvitge Biomedical Research Institute (IDIBELL), using the CytoTune^®^-iPS 2.0 Sendai Reprogramming Kit (Invitrogen, Carlsbad, CA, USA). Cells were passaged every 3–4 days at approximately 80% confluence. Colonies (1–3 mm diameter) were composed of uniformly sized cells, with a large and flat polygonal shape, smooth edges and high nucleus/cytoplasm ratio. The cells were named Ct PBiPS1-SV4F-1 (CTX1) and Ct PBiPS2-SV4F-1 (CTX2). As control lines, we used FiPS CTRL1-SV4F-7 (CTRL1) and FiPS CTRL2-SV4F-1 (CTRL2) cells. The registration and characterization of these hiPSCs can be found at https://www.isciii.es/QueHacemos/Servicios/BIOBANCOS/BNLC/Paginas/LineasiPS.aspx (accessed on 20 December 2022). Undifferentiated hiPSCs were maintained on hESC-qualified Matrigel^®^ (Corning Corp., Tewksbury, MA, USA)-coated 6-well plates in mTeSR Plus medium (StemCell Technologies, Vancouver, BC, Canada) at 37 °C and 5% CO_2_. Both hiPSCs (CTX1 and CTX2) showed normal karyotype and were contamination free.

### 2.3. Genetic Testing of iPSCs

DNA was extracted from all hiPSC lines using an automated genomic DNA purification kit (QIAsymphony SP^®^; Qiagen, Hilden, Germany). Coding exons and intronic boundaries of 251 genes related to inherited cardiovascular diseases and sudden cardiac death, including 121 genes associated with dilated cardiomyopathy (DCM) (Appendix A), were captured using a custom probe library (SureSelect Target Enrichment Kit for Illumina paired-end multiplexed sequencing method; Agilent Technologies, Santa Clara, CA, USA) and sequenced on the NovaSeq6000 platform (Illumina, San Diego, CA, USA). The read depth was greater than 30× (mean 250–400×). Exons that did not fulfill this standard were complementarily sequenced using the Sanger method. Likely protein-altering variants (missense, in-frame insertions/deletions, frameshift, nonsense and consensus splice site mutations) were evaluated, and bioinformatics analysis was performed using a custom pipeline including software for variant calling, genotyping and annotation. Variant frequency in the general population was extracted from the gnomAD database version r2.0, August 2017 (http://gnomad.broadinstitute.org accessed on 16 December 2022) [27]. We defined candidate variants as those with a minor allele frequency of <0.2% in any non-founder population of gnomAD. To establish the pathogenicity of identified candidate variants, we used a customized classification scheme based on the recommendations of the American College of Medical Genetics and Genomics [28]; the final classification of each variant was agreed upon by consensus between two cardiogeneticists with experience in interpretation of genetic variants.

### 2.4. Induction of Cardiac Differentiation

To differentiate iPSC-CMs, we followed a protocol adapted from Kleinsorge et al. [29]. Cells were plated on 6-well plates coated with Matrigel^®^ in mTeSR Plus medium at 200,000 cells/well. Differentiation was started when hiPSCs reached 70–80% confluence (1–2 days from plating). The mTeSR Plus medium was changed to RPMI 1640 (ThermoFisher Scientific, Waltham, MA, USA) supplemented with B-27 without insulin (ThermoFisher Scientific) and 50 mg/mL L-ascorbic acid 2-phosphate (Sigma-Aldrich, St. Louis, MO, USA) (RPMI/B-27 without insulin). The WNT pathway activator CHIR999021 (6 µM; StemCell Technologies) was added to the cells from day 0 to 2, followed by the addition of the WNT pathway inhibitor IWP-4 (5 µM; StemCell Technologies) from day 2 to 4. From day 8 to 11 the medium was changed for RPMI 1640 supplemented with B-27 with insulin (ThermoFisher Scientific) and 0.2 mg/mL L-ascorbic acid 2-phosphate (RPMI/B-27 with insulin). From day 11 to 16, we performed metabolic selection by changing the medium to RPMI 1640 without glucose (ThermoFisher Scientific) supplemented with B-27 with insulin, 50 mg/mL L-ascorbic acid and 4 mM L-lactate (Sigma-Aldrich). The medium was changed again to RPMI/B-27 with insulin from day 16 to 18 to allow the recovery of the cardiomyocytes, and, on day 18, hiPSC-CMs were digested with trypsin/EDTA (ThermoFisher Scientific) and replated at a lower density (1 million cells/mL) in RPMI/B-27 with insulin and 20% fetal bovine serum (FBS; ThermoFisher Scientific). From day 20 onwards hiPSC-CMs were maintained in RPMI/B-27 with insulin, and experiments were performed from day 24 to 30.

### 2.5. Immunostaining and Flow Cytometry

The expression of cardiac markers was analyzed by flow cytometry to determine the percentage of hiPSC-CMs. After differentiation, cells were dissociated with trypsin/EDTA, fixed with 4% paraformaldehyde (PFA) for 15 min at 4 °C, washed twice with PBS and blocked and permeabilized with a blocking and permeabilization solution composed of 5% FBS and 0.01% Triton-X100 (Sigma-Aldrich) in PBS for 30 min at room temperature (RT). Cells were then incubated with antibodies against myosin heavy chain (MHC; 1/200; 564408, BD Pharmingen, San Diego, CA, USA) and cardiac troponin I (cTnI; 1/200; 554409, BD Pharmingen) for 1 h at RT and then washed twice with PBS. Cells were resuspended in PBS and analyzed on a FACS Canto II cytometer (BD Bioscience, San Diego, CA, USA). Control isotypes (PE Isotype control, 550617; Alexa Fluor^®^ 647 Isotype control, 558713; both from BD Pharmingen) were used at the same concentration as the specific antibodies. Data were analyzed using FlowJo X (TreeStar Inc., Ashland, OR, USA).

### 2.6. Immunocytochemistry

For immunofluorescence analysis, hiPSC-CMs were seeded on coverslips, fixed with 4% PFA for 15 min at 4 °C and washed three times with PBS. The cells then were then blocked and permeabilized for 1 h at RT and incubated with the following primary antibodies: anti-α-sarcomeric actinin (SAA; 1/200; A7811, Sigma-Aldrich), anti-cTnT (1/200; ab8295, Abcam, Cambridge, UK) or anti-phospho-Histone γ-H2AX (1/200; 05-636-200, Merck Millipore, Burlington, MA, USA) overnight at 4 °C. Thereafter, the cells were washed three times with PBS and incubated with the secondary antibodies anti-mouse Alexa Fluor^®^ 488 (1/200; A11001, ThermoFisher Scientific) or anti-rabbit Alexa Fluor^®^ 555 (1/200; A21428, ThermoFisher Scientific) for 2 h at RT. The cells were then counterstained with 200 ng/mL DAPI (ThermoFisher Scientific) for 10 min and mounted with FluorSave^TM^ reagent (CalbioChem, San Diego, CA, USA). hiPSC-CMs were imaged with a Leica DM2500 fluorescence microscope (Leica Microsystems, Wetzlar, Germany). To assess sarcomeric disorganization, we utilized images of SAA staining, which specifically labels sarcomeres in muscle fibers. These images were processed and analyzed using the ImageJ software (NIH, Bethesda, MD, USA). First, the SAA staining images were transformed into a binary format, where sarcomeres appeared as black particles against a white background. This preprocessing step facilitated further analysis. Next, we quantified two key parameters, the organized area (OA) and the total area (TA) of the sarcomeres. The OA represents the area of regularly spaced Z-disks, which indicates organized sarcomeres. The TA encompasses the overall area of the sarcomeres, regardless of their regularity. Using the “analyze particles” tool in ImageJ, we measured the total area of particles falling within specific size and circularity index ranges for both OA and TA. For OA, the range was 5–40 µm² for particle size and 0.00–0.80 for circularity index. For TA, the range was 0–40 µm² for particle size and 0.00–1.00 for circularity index. After obtaining these measurements, we calculated the percentage of sarcomere disorganization. This was achieved by applying the formula (1 − OA/TA) ∙ 100, which quantifies the degree of disorganization based on a comparison between the organized area and the total area of the sarcomeres. A higher percentage indicates greater disorganization.

### 2.7. mRNA Analysis

RNA was isolated using the RNeasy Plus Mini Kit (Qiagen, Westburg BV, Leusden, The Netherlands), and cDNA was obtained by reverse transcription employing the PrimeScript RT Reagent Kit (Takara, Kusatsu, Japan). RT-qPCR was performed with the respective primers and TB Green^®^ Premix Ex Taq^TM^ (Tli RNase H Plus; Takara) on a ViiA 7 Real-Time PCR System (Applied Biosystems, Foster City, CA, USA).

### 2.8. Cell Viability

Cardiomyocyte damage was induced in hiPSC-CMs with different concentrations of DOX for 48 h. Cell viability was then determined using the Cell Counting Kit-8 (CCK8; Merck KGaA, Darmstadt, Germany). Cells were incubated with 10% CCK8 reagent for 3 h at 37 °C, and absorbance was measured at 450 nm using a microplate reader (Halo LED 8; Dynamica Scientific, Newport Pagnell, UK).

### 2.9. Lactate Dehydrogenase Assay

Supernatant fractions of hiPSC-CMs were assayed for lactate dehydrogenase (LDH) using the Cytotoxicity Detection Kit Plus (Roche, Indianapolis, IN, USA).

### 2.10. Annexin-V Staining

Cells were detached with trypsin/EDTA, centrifuged at 1000 rpm for 5 min, resuspended in binding buffer with annexin-V-FITC and DAPI (R&D Systems, Minneapolis, MN, USA) and incubated for 15 min at RT. The percentage of annexin-V-positive cells was measured by flow cytometry on a FACS Canto II cytometer.

### 2.11. Caspase 3/7 Activity Assay

Caspase 3/7 activity was measured using the Caspase-Glo^®^ 3/7 assay system (Promega, Madison, WI, USA). Briefly, the Caspase-Glo^®^ reagent was added to hiPSC-CMs seeded in 96-well plates and incubated for 30 min at RT. Luminescence was recorded on a Synergy H1 microplate reader (BioTek, Winooski, VT, USA).

### 2.12. Western Blotting

hiPSC-CMs were lysed in RIPA buffer supplemented with protease and phosphatase inhibitors (all from Sigma-Aldrich). Equal amounts of samples were mixed with non-reducing Laemmli sample buffer (BioRad, Hercules, CA, USA) and denatured at 96 °C for 5 min. Proteins were then separated on 10–12% SDS-polyacrylamide gels and transferred to polyvinylidene difluoride membranes (Immobilon-P; Millipore). Membranes were blocked with 5% non-fat dry milk powder in Tris-buffered saline. Primary antibodies used for western blotting were as follows: anti-tubulin (1/1000; T5168, Sigma-Aldrich), anti-caspase-3 (1/500; #9662, Cell Signaling Technology, Danvers, MA, USA), anti-cleaved caspase-3 (1/200; #9661, Cell Signaling Technology), anti-Bax (1/500; #5023, Cell Signaling Technology), anti-Bcl-2 (1/200; #4223, Cell Signaling Technology) and anti-phospho-Histone γ-H2AX (1/1000; 05-636-200, Merck Millipore). Secondary antibodies were anti-IgG rabbit (1/5000; P0448 Dako, Santa Clara, CA, USA) and anti-IgG mouse (1/5000; A9044, Sigma-Aldrich). Detection was carried out using peroxidase-conjugated antibodies and SuperSignal^TM^ West Femto (ThermoFisher Scientific). Reactions were visualized using an Amershan Imager 600 (GE Healthcare, Chicago, IL, USA) and quantified with the ImageJ software (ver. 1.53t, NIH, Bethesda, MD, USA).

### 2.13. Oxidative Stress Assessment

hiPSC-CMs were stained with either 5 µM DCFH-DA (2′,7′-dichlorofluorescin diacetate; Sigma-Aldrich) for 30 min, for whole-cell reactive oxygen detection, or 5 µM MitoSOX^TM^ Red (ThermoFisher Scientific) for 30 min, for mitochondrial superoxide detection. Cardiomyocytes were washed three times with PBS, detached with Trypsin/EDTA and analyzed on a FACS Canto II cytometer. Data were analyzed using FlowJo X.

For reduced glutathione (GSH) quantification, we used the GSH-Glo^®^ Glutathione Assay (Promega). Luminescence was measured on a Synergy H1 microplate reader (Agilent).

### 2.14. Mitochondrial Membrane Potential

hiPSC-CMs were seeded in 48-well plates and, after DOX treatment, were stained with 10 µg/mL JC-1 (T3168; ThermoFisher Scientific) for 20 min at 37 °C. Thereafter, the cells were washed with PBS, and the medium was changed for RPMI/B-27 with insulin. Cells were imaged using a Leica DMi8 Platform live cell microscope with controlled CO_2_ and temperature (Leica Microsystems). Mean fluorescence intensity was quantified using ImageJ.

### 2.15. Optical Mapping Acquisition and Image Processing

Functional changes in the electrophysiological behavior of hiPSC-CMs in the basal state and after DOX exposure were measured by optical mapping using macroscopic calcium transient analysis. Details of the protocols used for staining have been described previously [30]. Briefly, cells were stained with the fluorescent Ca^2+^ indicator Rhod-2 AM (Abcam, Cambridge, UK). A green light-emitting diode (LED) was used to illuminate the samples: LED CBT-90-G (peak power output 58 W; peak wavelength 524 nm; Luminus Devices, Billerica, MA) with a plano-convex lens (LA1951; focal length 25.4 mm; Thorlabs, Newton, NJ) and a green excitation filter (D540/25X; Chroma Technology, Rockingham, VT, USA) [31]. To capture the light emitted by calcium transients in the samples, a CMOS-sensor Camera system (2400 × 2400 pixel; Kinetix Scientific, Auckland, New Zealand) was utilized along with a custom emission filter (ET585/50–800/200M; Chroma Technology) designed for rhod 2 emission. This emission filter was positioned in front of a high-speed camera lens.

The complexity of the arrhythmias was assessed using optical mapping. The number of re-entries in the propagation process was used as a surrogate for arrhythmia complexity, as previously described [31]. Custom MATLAB software (MATLAB 9.11, MathWorks, Inc., Natick, MA, USA), adapted from the methodology presented by Fambuena-Santos et al. [26], was used to identify phase singularities (PS) in the phase maps. The PS detection utilized the topological charge algorithm by Bray and Wikswo [32]. The complexity of the arrhythmia was then computed as the number of phase singularities per square centimeter in the standardized video format. The activation frequency was estimated as the maximum peak in the power spectral density of the temporal signals. Only peaks within the frequency range of 0.1–12 Hz were considered as candidates for the activation frequency. The power spectral density was calculated using Welch’s periodogram [33] with a window length of 3 sec and a 1 sec overlap between windows. Finally, the mean activation frequency was computed for the entire cell culture.

### 2.16. miRNA Analysis by RT-qPCR

miRNA was isolated using the miRNeasy^®^ Mini Kit, reverse transcription was performed with the miRCURY LNA Universal RT miRNA PCR Kit, and RT-qPCR was performed with the respective primers and the miRCURY LNA SYBR Green PCR Kit (all from Qiagen). miRNAs were quantified using a Viia TM 7 Real System, and the results were analyzed with QuantStudio Real-Time PCR Software (v1.4.3, Applied Biosystems).

### 2.17. Statistical Analysis

Quantitative variables are presented as mean ± standard deviation. Groups were compared with one-way ANOVA and post hoc analysis. The PS density and mean activation frequency were compared between the control and cardiotoxicity lines using the Mann–Whitney U test. Analyses were conducted with the GraphPad Prism 8^®^ software (ver. 8.0.2 San Diego, CA, USA). A *p*-value of <0.05 was considered significant with a 95% confidence interval.

## 3. Results

### 3.1. Donors

We generated hiPSCs from pediatric oncology patients with cardiotoxicity. We recruited 2 patients at 2 clinical sites who were 4 and 14 years old when they experienced anthracycline-induced acute cardiotoxicity. Control cell lines derived from healthy subjects were matched for age and sex as much as possible (Table 1).

### 3.2. Generation of hiPSCs from Pediatric Oncology Patients

The hiPSCs from patient donors (CTX1 and CTX2) were generated from PBMCs. As control lines, we used two hiPSC lines (CTRL1 and CTRL2) generated from foreskin/scrotum fibroblasts of two healthy subjects (Table 1 and https://www.isciii.es/QueHacemos/Servicios/BIOBANCOS/BNLC/Paginas/LineasiPS.aspx (accessed on 20 December 2022)). Several iPSC clones were generated from donors, and each was validated for pluripotency markers and teratoma formation and screened for chromosomal abnormalities (results not shown).

The four iPSC cell lines were screened for known genetic variants of DCM, as it is known that patients harboring these disease-causing variants can develop more severe and earlier phenotypes after a “second hit” such as anthracycline treatment. We performed next-generation sequencing of 251 genes curated and associated with DCM. The list of genes and their candidate variants identified are detailed in Appendix A. No disease-causing (pathogenic/likely pathogenic) variants were identified in CTRL or CTX hiPSCs. Of the 12 identified variants, 4 corresponded to variants of uncertain significance (VUS), all from the CTRL cells, and the remaining variants were benign/likely benign.

### 3.3. Differentiation of iPSCs into Ventricular Cardiomyocytes

Cardiac differentiation was performed by modulating the WNT signaling pathway, as described (Figure 1A) [29]. Beating cardiomyocytes could be observed from day 8 of differentiation onwards (Figure 1B).

To determine the cardiomyocyte differentiation yield, we analyzed the expression of two cardiac markers, MHC and cTnI, observing that >90% of the cells were positive for both (Figure 1C). Supporting this finding, immunostaining for two sarcomeric proteins (SAA and cTnT) revealed a typical striated sarcomere pattern in many cells (Figure 1D). Finally, we analyzed the expression of two gene markers of pluripotency (*OCT4* and *SOX2*) and two gene markers of cardiomyocytes (*MYH7* and *TNNT2*) along the cardiac differentiation process (Figure 1E). As expected, high expression of *OCT4* and *SOX2* was evident at the start of differentiation, with no expression of *MYH7* and *TNNT2*. From day 5 of differentiation onwards, however, *OCT4* and *SOX2* expression disappeared, concomitant with the expression of *TNNT2* and *MYH7* by day 11; both genes were highly expressed by day 18 of differentiation, when hiPSC-CMs were replated for further experiments.

### 3.4. Analysis of DOX-Induced Cytotoxicity and Apoptosis in hiPSC-CMs

We next assessed the cell viability of cultured CTRL and CTX iPSC-CMs, finding no differences between groups in the basal state, as measured by CCK8 and LDH analysis, annexin-V staining, caspase 3/7 activity and Bcl2/Bax ratio (Figure 2). To characterize the cardiotoxicity induced by DOX, we first tested different concentrations of DOX in iPSC-CM cultures for 48 h and assessed cell viability. We found that starting from a dose of 1 µM of DOX, CTX hiPSC-CMs were significantly more sensitive to DOX than equivalent CTRL hiPSC-CMs (Figure 2A), as assessed with CCK8. Moreover, the half-maximal inhibitory concentration (IC_50_) of DOX was significantly higher in CTRL hiPSC-CMs than in CTX hiPSC-CMs (2.26 µM vs. 0.9 µM, respectively; Figure 2B). In accordance with these findings, LDH activity in the culture supernatants of cells treated with 1 µM of DOX was significantly higher in CTX hiPSC-CM cultures than in CTRL hiPSC-CMs (Figure 2C), and phase contrast microscopy images of DOX-treated CTRL and CTX hiPSC-CM cultures showed higher numbers of dead cells in the latter (Figure 2D). Analysis of annexin-V staining by flow cytometry revealed that the percentage of apoptotic cells was significantly higher in the CTX hiPSC-CM cultures treated with DOX than in the equivalent CTRL cultures (Figure 2E). Finally, DOX-treated CTX hiPSC-CMs exhibited higher caspase 3/7 activity (Figure 2F) as well as higher levels of caspase-3, cleaved caspase-3 and Bax/Bcl2 ratio than equivalent CTX hiPSC-CMs (Figure 2G–J). Overall, these results show that cardiomyocytes differentiated from iPSCs of patients with acute anthracycline toxicity show an enhanced sensitivity to DOX.

### 3.5. Effects of DOX on Oxidative Stress

As oxidative stress is a major contributor to DOX-induced cardiotoxicity, we determined whether the levels of ROS in hiPSC-CMs differed before and after DOX treatment. For whole-cell ROS detection, we utilized DCFH-DA staining. The results showed that while basal ROS levels were not different between CTRL and CTX hiPSC-CMs, ROS levels were significantly higher in DOX-treated CTX hiPSC-CMs than in equivalent CTRL hiPSC-CMs (Figure 3A). In addition, DOX treatment did not significantly increase ROS levels in CTX hiPSC-CMs, but did so in CTX hiPSC-CMs. Similar results were obtained after MitoSOX^®^ staining, which detects mitochondrial ROS (Figure 3B). Additionally, we examined the levels of reduced glutathione (GSH), the depletion of which indicates higher levels of ROS. The results showed that GSH levels were significantly lower in CTX hiPSC-CMs treated with DOX than in equivalent CTRL hiPSC-CMs (Figure 3C).

To gain a better understanding of the observed differences between CTRL and CTX cardiomyocytes, we investigated the expression of several genes related to ROS handling (Figure 3D). Prior to DOX treatment, we observed that the expression levels of *CAT*, which encodes the enzyme catalase responsible for peroxide hydrogen detoxification, were significantly lower in CTX hiPSC-CMs than in CTRL hiPSC-CMs. These findings suggest diminished antioxidative activity in the patient-specific cell cultures. Surprisingly, levels of NRF1, a transcription factor involved in the expression of several antioxidant genes, were higher in CTX hiPSC-CMs, but these differences were not observed after doxorubicin treatment. Following DOX treatment, levels of *CAT* remained lower in CTX hiPSC-CMs than in CTRL hiPSC-CMs. Analysis of *SOD1*, which encodes the enzyme superoxide dismutase 1, responsible for detoxifying superoxide, revealed similar levels in CTX and CTRL hiPSC-CMs but a decrease in CTX cultures after DOX treatment. No changes were seen in *NRF2* levels across cells or treatments. In contrast to the evident gene expression changes, the protein expression levels of CAT and SOD1 before DOX treatment were unchanged as assessed by Western blotting (Figure 3E–G), which is possibly due to the sensitivity of the technique. Nonetheless, we observed that SOD1 levels were lower in CTX hiPSC-CMs than in CTRL hiPSC-CMs after DOX treatment. Notably, the expression levels of *NQO2*, a thioredoxin involved in detoxification pathways, were higher in CTX hiPSC-CMs than in CTRL hiPSC-CMs prior to DOX treatment (Figure 3H), possibly indicating compensatory mechanisms. Similarly, the expression of endothelial nitric oxide synthase (*NOS3*), which can exert antioxidant activities, was higher in CTX hiPSC-CMs than in CTRL hiPSC-CMs before DOX treatment (Figure 3I). However, these differences were lost after DOX exposure.

### 3.6. DOX-Induced Genotoxic and Mitochondrial Damage in hiPSC-CMs

It is well established that one of the primary mechanisms by which DOX induces damage in cardiomyocytes is through binding to DNA topoisomerase-IIβ (TOP2B), resulting in double-strand DNA breaks [34]. These breaks impair transcription, leading to mitochondrial damage and ROS formation. We investigated DOX-induced genotoxic damage using immunofluorescence analysis of the phosphorylated histone γ-H2AX, a marker of double-strand DNA breaks. We found a greater proportion of nuclear phosphorylated histone γ-H2AX staining in CTX hiPSC-CMs treated with DOX than in equivalent CTRL hiPSC-CMs (Figure 4A). This was corroborated by the quantification of phosphorylated histone γ-H2AX protein levels (Figure 4B,C).

We next tested whether DOX treatment induces mitochondrial damage in hiPSC-CMs by assessing JC-1 dye staining before and after DOX exposure. JC-1 functions as a mitochondrial potential sensor by accumulating in the mitochondria and forming aggregates that emit red fluorescence (~590 nm). When mitochondria become damaged and depolarized, JC-1 fails to accumulate in the mitochondria and instead emits green fluorescence (~529 nm), resulting in a decrease in the red/green fluorescence intensity ratio. While no differences were observed in the JC-1 levels of CTRL and CTX hiPSC-CMs in the basal state, DOX treatment induced a significant reduction in this intensity ratio in CTX hiPSC-CMs, indicating reduced mitochondrial depolarization (Figure 4D,E).

### 3.7. Sarcomere Disassembly and Gene Expression in hiPSC-CMs

DOX triggers sarcomere disorganization and disassembly in cardiomyocytes, impairing their contractive functional properties. To study the effect of DOX in the arrangement of sarcomeres, we immunostained cells for sarcomeric α-actinin and cTnI (Figure 5A,B). The results showed an increase in sarcomere disorganization after DOX treatment both in CTRL and CTX hiPSC-CMs, although it was significantly greater in the latter.

We next investigated differences in the expression of a panel of genes encoding sarcomere proteins that could potentially account for the sarcomere disassembly. Ankyrin 2 (*ANK2*), which is located in the Z-discs and plays a role in the stabilization of actinin and obscurin, and myosin light chain 6 (*MYL6*), a sarcomeric protein involved in regulating the activity of cardiac myosin, were the only genes whose expression was lower in CTX hiPSC-CMs prior to DOX treatment (Figure 5D,E). Conversely, levels of cardiac α-actin *(ACTC1)* (Figure 5F), myosin light chain 3 (*MYL3*) (Figure 5G) and troponin I3 (*TNNI3*) (Figure 5I) were higher in CTX hiPSC-CMs. Following DOX treatment, both CTRL and CTX hiPSC-CMs showed decreases in the expression of *ANK2*, *MYH6*, *ACTC1* (Figure 5D–F), *MYH7* (Figure 5H), *TNNI3*, (Figure 5I) and nebulette (*NEBL)* (Figure 5J). *MYL3* expression was unaffected by DOX treatment (Figure 5G), and no differences in the expression of titin (*TTN*) or troponin T2 (*TNNT2*) were found, irrespective of cell type or treatment (Figure 5K,L).

### 3.8. TGF-β Signaling in hiPSC-CMs

Ventricular cardiomyocytes are capable of activating inflammatory and fibrogenic programs in response to stress [35]. We therefore studied the expression of genes related to fibrosis and TGF-β signaling in hiPSC-CMs before and after DOX treatment. Of note, the expression of Transforming Growth Factor Beta Receptor 1(*TGFBR1*), a type I transmembrane receptor protein involved in TGF-β signaling pathways, and mothers against decapentaplegic homologs 2 and 4 (*SMAD2* and *SMAD4,* respectively), was significantly higher in CTX hiPSC-CMs than in CTRL hiPSC-CMs before DOX treatment (Figure 6A–C). It is worth mentioning that SMAD2 and SMAD4 are critical components of the TGF-β signaling pathway. SMAD2 is phosphorylated and activated by TGFBR1, translocating to the nucleus where it regulates the transcription of target genes involved in cellular processes such as proliferation and differentiation. We found that DOX treatment did not reverse the dysregulated expression of TGF-β pathway genes (*TGFBR1*, *SMAD2* and *SMAD4*) in CTX hiPSC-CMs. Additionally, we examined the expression of periostin (*POSTN*) and fibrillin 1 (*FBN1*), which are implicated in cardiac fibrosis (Figure 6D,E). We observed that *POSTN* expression was higher in CTX hiPSC-CMs than in CTRL hiPSC-CMs both before and after DOX treatment. Treatment with DOX did not affect the expression of *FBN1* in CTRL hiPSC-CMs; however, it induced the expression of *FBN1* in CTX hiPSC-CMs. Furthermore, the levels of *FBN1* in CTX hiPSC-CMs treated with DOX were significantly higher than those in equivalent CTRL hiPSC-CMs. The elevated expression of fibrosis-related genes suggests that the TGF-β signaling pathway could be hyperactivated in the cardiomyocytes of patients with cardiotoxicity.

### 3.9. Differences in Arrhythmogenicity between CTRL and CTX hiPSC-CMs Treated with DOX

To assess whether the disparities in the expression of pro-fibrotic genes could be reflected in a higher susceptibility to DOX in terms of arrhythmia complexity, we performed optical mapping of CTRL and CTX hiPSC-CMs treated or not with DOX to quantify differences in fibrillatory patterns. A total of 78 videos from 30 independent samples (2.6 ± 0.8 videos per sample) were analyzed. Table 2 summarizes the number of recordings performed under basal and DOX treatments. The number of independent differentiation experiments is reflected in the parentheses.

The results showed that CTX hiPSC-CMs had a higher susceptibility to DOX in terms of arrhythmia complexity. An illustrative example of the accentuated complexity is shown in Figure 7A, with four distinct fibrillatory patterns observed in both CTRL and CTX hiPSC-CMs under basal and DOX conditions. In the first two patterns, which belong to a CTRL hiPSC-CM line, only a limited number of re-entries driving fibrillation were observed. Upon addition of DOX to CTRL cells (second image from the left), there was no significant augmentation in the number of wavefronts or re-entries, and only one rotor positioned in the upper section of the image appeared to be driving the fibrillation. The same conditions are depicted for the CTX hiPSC-CMs in the subsequent two images, which revealed two interrelated phenomena. First, wavefronts in pattern 3 under basal conditions were larger than those observed during DOX treatment. Second, DOX administration introduced a multitude of new wavefronts and re-entries (fourth image) that were absent during basal conditions (third image).

The mean activation frequency of CTRL and CTX hiPSC-CMs was unchanged by DOX treatment, and so the different registers with or without the drug were represented together for each cell line. We found that the mean activation frequency was significantly higher in CTX hiPSC-CMs than in CTRL hiPSC-CMs (Figure 7B). Moreover, CTX hiPSC-CMs exhibited heightened susceptibility to DOX, and the average number of phase singularities per square centimeter (PS/cm^2^) increased significantly after DOX treatment. In the context of cardiac arrhythmias, PS/cm^2^ refers to points in the cardiac tissue where the electrical activation is disorganized or irregular. These singularities are indicative of the presence of wavefront collisions and re-entries, which are common phenomena associated with arrhythmias. The PS/cm^2^ values calculated for both lines and conditions are represented in Figure 7C. The results showed that CTRL cells did not have significantly different PS densities (45.2 ± 19.6 vs. 49.6 ± 7.9, *p* = 0.29 in CTRL1 and 24.6 ± 27.6 vs. 33.5 ± 18.7, *p* = 0.44 in CTRL2). By contrast, CTX hiPSC-CMs showed a significant increase in PS density when treated with DOX (32.6 ± 47.9 vs. 99.2 ± 22.5, *p* = 6.3·10-4 in CTX1 and 11.3 ± 15.4 vs. 48.3 ± 49.7, *p* = 0.006 in CTX2). The heightened complexity of arrhythmias observed in CTX hiPSC-CMs suggests an increased susceptibility to cardiac adverse events, including arrhythmias, experienced by patients with cardiotoxicity due to anthracycline treatment.

To deepen our understanding of these phenomena, we wanted to investigate whether alterations in gene expression, specifically those encoding ion channels (as depicted in Figure 7D), could contribute to these observed differences. We focused on examining the expression levels of Sarco(endo)plasmic reticulum Ca2+ ATPase 2 (*SERCA2*) and cardiac ryanodine receptor 2 (*RYR2*), which are essential for regulating calcium handling in cardiomyocytes during excitation/contraction coupling.

Reduced expression of both *SERCA2* and *RYR2* was detected in CTX hiPSC-CMs compared to CTRL hiPSC-CMs after DOX treatment. These findings suggest potential abnormalities in calcium dynamics within CTX hiPSC-CMs. Furthermore, we assessed the expression of the cardiac sodium voltage-gated channel alpha subunit 5 (*SCN5A*) and found that levels were significantly higher in CTX hiPSC-CMs compared to CTRL hiPSC-CMs, both at basal conditions and after DOX treatment. In addition, Connexin 43 (*Cx43*), a gap-junction protein critical for efficient electrical conduction in the heart, exhibited significantly lower expression levels in CTX hiPSC-CMs compared to CTRL hiPSC-CMs after DOX treatment. Impaired functioning of these channels can disrupt normal electrical signaling in the heart, potentially leading to early afterdepolarizations or delayed afterdepolarizations, which are known arrhythmogenic phenomena.

### 3.10. miRNA Expression Analysis in hiPSC-CMs

Some miRNAs have been shown to have cardioprotective effects in different cardiac diseases [36,37,38]. For this reason, we examined the expression of several cardioprotective miRNAs in CTRL and CTX hiPSC-CMs before and after DOX treatment (Figure 8A–G). No differences were found in the levels of hsa-miR-1-3p, hsa-miR-133a-3p, hsa-miR-148a-3p and hsa-miR-150-5p between CTRL and CTX hiPSC-CMs before DOX treatment. However, the levels of hsa-miR-22-3p, hsa-miR-30b-5p, hsa-miR-90-3p and hsa-miR-4732-3p were significantly lower in CTX hiPSC-CMs than in CTRL hiPSC-CMs before adding the drug. Notably, the levels of these miRNAs were upregulated in CTRL hiPSC-CMs after DOX treatment, consistent with previous reports [39], but not in CTX hiPSC-CMs. Overall, these results suggest that there may be an inherent dysregulation of these miRNAs in CTX hiPSC-CMs, which could contribute to their compromised cardioprotective response. These findings suggest that the expression and/or regulation of these miRNAs might play a crucial role in cardioprotection against DOX toxicity and that their dysregulation in CTX hiPSC-CMs might contribute to the enhanced susceptibility of these cells to DOX-induced cardiotoxicity.

## 4. Discussion

In the present study, we generated two hiPSC lines from pediatric oncology patients with acute cardiotoxicity after anthracycline treatment to evaluate the underlying mechanisms. To exclude a genetic basis for the predisposition to cardiotoxicity, we first investigated potential genetic variants associated with DCM in these cells using a panel of 121 genes. No disease-causing variants were identified in either the CTX or CTRL groups, and no significant differences were found in the number of VUS between groups that could account for susceptibility to cardiotoxicity. These results indicate that the hiPSC lines did not harbor disease-causing variants associated with DCM, supporting their suitability for further investigation in the context of potential interactions with other factors such as anthracycline treatment. hiPSCs were differentiated to cardiomyocytes to develop a cellular model of DOX-induced cardiotoxicity. Our first observation was that cell viability after DOX exposure was significantly poorer in CTX hiPSC-CMs than in CTRL hiPSC-CMs. Similar results were found for other measures of cell viability. These findings agree with a study by Burridge et al. [23], who showed that hiPSC-CMs from patients with breast cancer and anthracycline-induced cardiotoxicity were more vulnerable to DOX treatment in vitro than control cardiomyocytes. Our results also accord with a study by Kitani et al. [24], showing that cardiomyocytes from patients with cardiac dysfunction after trastuzumab treatment were more susceptible to trastuzumab treatment in vitro.

To understand how DOX triggered greater cell death in CTX hiPSC-CMs, we studied the levels of apoptosis effector proteins. Caspase-3 activation by DOX is well-characterized [40,41,42], and we found that cleaved caspase-3 levels were enhanced in DOX-exposed CTX hiPSC-CMs, concomitant with a higher ratio between the pro-apoptotic protein Bax and the anti-apoptotic protein Bcl-2. DOX treatment also causes oxidative damage [43], and we observed that both total levels of ROS and mitochondrial superoxide were higher in DOX-exposed CTX hiPSC-CMs than in CTRL cells. This was accompanied by reduced levels of GSH, which is related to higher levels of oxidative stress [44], and with changes in the expression of genes related to ROS detoxification. For example, *SOD1* levels were maintained in CTRL hiPSC-CMs exposed to DOX, whereas a decrease was seen in CTX hiPSC-CMs after DOX treatment. SOD1 converts superoxide to hydrogen peroxide and oxygen, which maintains redox homeostasis in the cell [45]. These differences were partly confirmed by Western blotting. We also observed differences in the expression of CAT, which transforms hydrogen peroxide into oxygen and water [46]. CAT levels were lower in CTX hiPSC-CMs than in CTRL hiPSC-CMs both before and after DOX exposure, which might reflect a pre-existing defect in oxidative metabolism, making the cells more susceptible to the effects of DOX. Contrastingly, *NQO2* levels were higher in CTX hiPSC-CMs than in CTRL hiPSC-CMs, but decreased after DOX treatment. *NQO2* encodes NRH:quinone oxidoreductase 2, which detoxifies quinones and protect cells from oxidative stress [47], and thus higher levels could be protective against oxidative stress. Indeed, the higher levels of this gene in CTX hiPSC-CMs before DOX exposure could represent an adaptive response to the disease state, which might provide a level of protection against oxidative stress. We also observed that *NOS3*, which encodes nitric oxide synthase 3, was expressed at a higher level in CTX hiPSC-CMs than in CTRL hiPSC-CMs under basal conditions. This is interesting, as the genomic variant rs1799983 of *NOS3* has a protective effect against DOX in at-risk patients [48]. Furthermore, mice with knock-out of *NOS3* show increased protection against DOX damage [49]. Nevertheless, no significant differences were observed in *NOS3* expression between cells after DOX treatment. This has previously been reported after DOX treatment of CTX hiPSC-CMs derived from breast cancer patients with cardiotoxicity [23], pointing to a conserved response of this cell type to DOX regardless of the type of cancer and the use of iPSCs derived from adult or pediatric patients with anthracycline-induced cardiotoxicity. Levels of *NRF1* were higher in CTX hiPSC-CMs than in CTRL hiPSC-CMs, which could be indicative of a compensatory response, but there was loss of expression of this gene following DOX treatment. This gene encodes transcriptional factors that enhance the expression of antioxidant genes [50]; thus, higher levels of *NRF1* exert a protective role against oxidative stress. Overall, the interpretation of ROS data is complex and involves several limitations. For instance, (i) both the qPCR and Western blot methods provide indirect measurements of oxidative stress, rather than directly quantifying the oxidative stress itself; (ii) oxidative stress is interconnected with various cellular processes, including antioxidant defense mechanisms, that can influence gene expression or protein levels; and (iii) oxidative stress is a dynamic process that can have transient or localized effects within cells, so measurements of the cellular state at specific time points do not reflect temporal or spatial variations in oxidative stress levels. These factors should be taken into account when interpreting results observed in our experiments.

Another intriguing aspect to explore is the impact of doxorubicin on genomic instability. In line with this, Zhang et al. reported that the genomic damage caused by DOX via the inhibition of Top2β had a crucial role in DOX-induced cardiotoxicity [11]. We thus studied the p-histone γ-H2AX as a marker of double-strand DNA breaks [51], finding that levels were higher in CTX hiPSC-CMs than in CTRL hiPSC-CMs after DOX exposure. We also examined mitochondrial damage by JC-1 staining, which revealed a less marked loss of membrane potential in CTX hiPSC-CMs after DOX exposure compared with CTX hiPSC-CMs. This indicates that CTX hiPSC-CMs had higher levels of DOX-induced genotoxic and mitochondrial damage, which supports the higher susceptibility to DOX observed previously.

DOX treatment induces sarcomeric disorganization and alters the expression of several sarcomeric proteins [23,52]. Immunostaining for cTnT and sarcomeric α-actinin revealed greater sarcomeric disarray in CTX hiPSC-CMs after DOX treatment, which was also reflected in the downregulation of *MYH7* and upregulation of *MYL3.* Interestingly, a recent study showed that Myl3 is a good cardiotoxicity biomarker [53], and so higher levels of *MYL3* might be related to cardiotoxicity induced by anthracyclines. We also observed higher levels of *ACTC1*, which encodes cardiac α-actinin, in CTX hiPSC-CMs in the basal state, but not after DOX treatment. Regarding the higher expression of *TNNI3* in CTX hiPSC-CMs than in CTRL cells before DOX treatment, this might be relevant because high levels of *TNNI3* can have adverse effects [54]. Likewise, *ANK2* levels were lower in CTX hiPSC-CMs than in CTRL hiPSC-CMs. Higher levels of *ANK2* could help to preserve the sarcomere structure, as ankyrin 2 binds the sarcomere to the sarcolemma, maintaining the sarcomere’s integrity during contraction [55]. Nonetheless, while we have not yet fully elucidated the underlying reasons for all the observed gene expression changes between the CRTL and CTX cell lines, it is evident that the inherent immaturity of hiPSC-CMs significantly contributes to these findings.

DOX has been reported to induce the differentiation of cardiac fibroblasts into myofibroblasts, causing heart interstitial fibrosis [56]. Indeed, cardiomyocytes can also activate inflammatory and fibrogenic programs in response to damage [35], which prompted us examine the expression of TGF-β signaling and pro-fibrotic genes. Our results revealed an upregulation of *TGFBR1*, *SMAD2*, *SMAD4*, *POSTN* and *FBN1* in CTX iPSC-CMs. Periostin, which is highly upregulated in fibrotic tissue, plays a critical role in cardiac hypertrophy and ventricular remodeling [57]. Similarly, it was reported that myofibroblasts, a key cell type involved in fibrotic responses following cardiac injury, are derived from diverse cardiac cell populations, including cardiomyocytes, and genetic lineage tracing of these cells can provide insights into their origin and function [58]. In the same line, Fibrillin 1 upregulation has been reported in the context of cardiac remodeling by ventricular tachypacing in a congestive heart failure canine model [59] and is associated with cardiac fibrosis [60,61]. These findings suggest that the upregulation of these genes may be involved in the development of cardiac fibrosis and dysfunction.

We also aimed to investigate the disparities in arrhythmia complexity between CTRL and CTX hiPSC-CMs treated with DOX and their potential association with the expression of pro-fibrotic genes. Optical mapping allowed us to quantify differences in fibrillatory patterns. CTX hiPSC-CMs demonstrated a greater susceptibility to DOX in terms of arrhythmogenicity, as the mean number of phase singularities (PS/cm^2^), indicating areas of disorganized or irregular electrical activation, was significantly increased. These findings suggest a relationship between the expression of pro-fibrotic genes and arrhythmia complexity, supporting the notion that pro-fibrotic gene dysregulation could contribute to the increased vulnerability to cardiac adverse events in the context of anthracycline-induced cardiotoxicity. Moreover, our results demonstrated altered expression of genes encoding crucial ion channels in CTX hiPSC-CMs, including SERCA2, RYR2, Cx43 and SCN5A. These findings suggest potential disruptions in calcium and sodium ion handling, as well as compromised electrical conduction, which could contribute to the observed differences in activation frequency and increased sensitivity to DOX treatment in CTX hiPSC-CMs compared to CTRL hiPSC-CMs.

Finally, we studied the expression of several miRNAs with known cardioprotective roles in several cardiac diseases. We and others have demonstrated that miR-1-3p and miR-133a-3p are both highly expressed in cardiomyocytes and their precursor cells [62,63,64], and that treatment with mimics of these miRNAs exerts cardioprotective effects against cardiac hypertrophy [63,64,65,66]. Other studies have shown that miR-30b-5p and miR-22-3p can contain fibrosis in mice after myocardial infarction [67], and that the overexpression of miR-30b-5p in the heart of transgenic mice reduces infarction size and cardiomyocyte cell death after ischemic/reperfusion injury [68]. With respect to miR-148a-3p, the transduction of this miRNA with adeno-associated viruses was found to protect the heart in a murine model of pressure overload, suppressing ventricular dilation [69], and the administration of M2 macrophage-derived exosomes enriched with miR-148a-3p diminished myocardial damage in a model of ischemic/reperfusion injury [37]. Similarly, miR-150-5p has antifibrotic effects [70,71] and its administration diminished cardiomyocyte death in a rat model of myocardium damage by septic shock [72] and in a murine model of ischemic/reperfusion injury [36].

We observed that the expression of hsa-miR-1-3p, hsa-miR-22-3p hsa-miR-30b-5p, hsa-miR-90-3p, has-miR133a-3p, hsa-miR-148a-3p and hsa-miR-150-5p was lower in CTX hiPSC-CMs than in CTRL hiPSC-CMs after DOX exposure. This is in line with studies showing that downregulation of these miRNAs is associated with cardiac fibrosis, adverse remodeling, cardiac hypertrophy [73], heart failure [74] and myocardial infarction. Indeed, all the miRNAs studied, with the exception of miR-4732-3p, showed an increase in expression in CTRL hiPSC-CMs after DOX treatment. The failure of cardiomyocytes derived from cardiotoxic patients to upregulate the expression levels of the aforementioned miRNAs may suggest a reduced capacity for adaptive response to DOX-induced damage. Indeed, downregulation of the miR-30 family was observed in both acute and chronic models of DOX-induced injury, highlighting its role in the pathogenesis of DOX-induced heart failure. It was demonstrated that miR-30 acts as a regulator of the β-adrenergic pathway, modulating the activity of β1- and β2-adrenoceptors and targeting target the pro-apoptotic gene BNIP3L/NIX, suggesting a potential anti-apoptotic role of miR-30 in cardiac cells [75]. miR-4732-3p expression was also lower in CTX hiPSC-CMs than in CTRL hiPSC-CMs before DOX treatment, which might explain the weaker resistance of CTX hiPSC-CMs to DOX. Indeed, we recently reported that miR-4732-3p is downregulated in the serum of breast cancer patients with cardiotoxicity induced by anthracyclines and has a cardioprotective role against DOX both in vitro and in vivo [76]. Furthermore, in contrast to the other miRNAs tested, miR-4732-3p expression was not stimulated by DOX in CTRL-hiPSC-CMs. miR-4732-3p is a primate-specific miRNA expressed not only in cardiac cells, but also in erythrocytes, where it was found to target critical components of the TGF-β pathway (SMAD2 and SMAD4) and repress SMAD2/4-dependent TGF-β signaling, and to promote cell proliferation during erythroid differentiation [77]. This functionality of miR-4732-3p could explain the higher levels of SMAD2 and SMAD4 in CTX iPSC-CMs. Another study revealed the dysregulation of microRNAs in blood from monozygotic twins discordant for congenital heart disease, with miR-4732-3p upregulated in such cases [78]. Additionally, circulating miR-4732-3p has emerged as a novel prognostic biomarker for acute heart failure (AHF), showing potential in predicting adverse outcomes and ventricular hypertrophy in affected patients [79]. Furthermore, we demonstrated that extracellular vesicles from mesenchymal stromal cells contain miR-4732-3p, which were identified as a hypoxia-regulated cargo, and exerted cardioprotective effects against ischemic insult both in vitro and in an experimental model of myocardial infarction [80]. These findings, collectively, highlight the significance of miR-4732-3p in cardiac health and suggest its potential as a therapeutic target or prognostic biomarker in anthracycline-induced cardiotoxicity, congenital heart disease and AHF. In line with these results, Goukassian et al. [81] explored the effects of short-duration low-Earth-orbit spaceflight on circulating plasma small extracellular vesicle microRNA expression, finding that miR-4732-3p is significantly upregulated up to 3 days post-landing following spaceflight as a result of the exposure of astronauts to various stressors during spaceflight, including microgravity and ionizing radiation, both linked to cardiovascular and neurodegenerative diseases. The upregulation of miR-4732-3p in response to cardiovascular stressors suggests its potential involvement in the adaptive responses and pathological processes associated with cardiovascular health during space travel. Thus, future studies are needed to investigate the downstream targets and pathways influenced by this miRNA to gain a comprehensive understanding of its functional role in cardiotoxicity.

## 5. Conclusions

We show that hiPSC-CMs obtained from pediatric oncology patients with anthracycline-induced cardiotoxicity have a higher susceptibility to DOX in vitro. The generation of a cellular model with naturally low miR-4732-3p expression and with increased susceptibility to cardiotoxic damage has potential value for studying the molecular mechanisms underlying this susceptibility and in the search for new therapeutic targets. Nonetheless, the present study has several limitations that should be addressed. First, the cardiotoxicity observed in the patients used to derive hiPSCs is idiopathic, meaning that it is not possible to obtain isogenic controls for the CTX hiPSC-CMs, which would be a better control to dissect the differences in the susceptibility to DOX. For this reason, it would be necessary to increase the number of CTRL and CTX lines studied. However, the cell lines were obtained from pediatric patients with acute cardiotoxicity, which has a very low incidence, making it difficult to increase the number of independent CTX hiPSC-CMs. Secondly, the cardiomyocytes obtained with the protocol used are immature and might not reflect the phenotype of mature cardiomyocytes against DOX damage. In this regard, the model of cardiotoxicity induced by DOX could be improved by obtaining mature cardiomyocytes, such as 3D models. Lastly, the cardiomyocytes used in this study are mainly ventricular. It would be necessary to study cardiotoxicity induced by anthracyclines also in atrial cardiomyocytes and even in other cell types such as cardiac fibroblasts or endothelial cells. Overall, the precise mechanisms underlying the observed changes in gene expression in patient-specific cardiomyocytes following DOX treatment are complex and multifactorial, and further research is needed to fully elucidate these links.

## Figures and Tables

**Figure 1 antioxidants-12-01378-f001:**
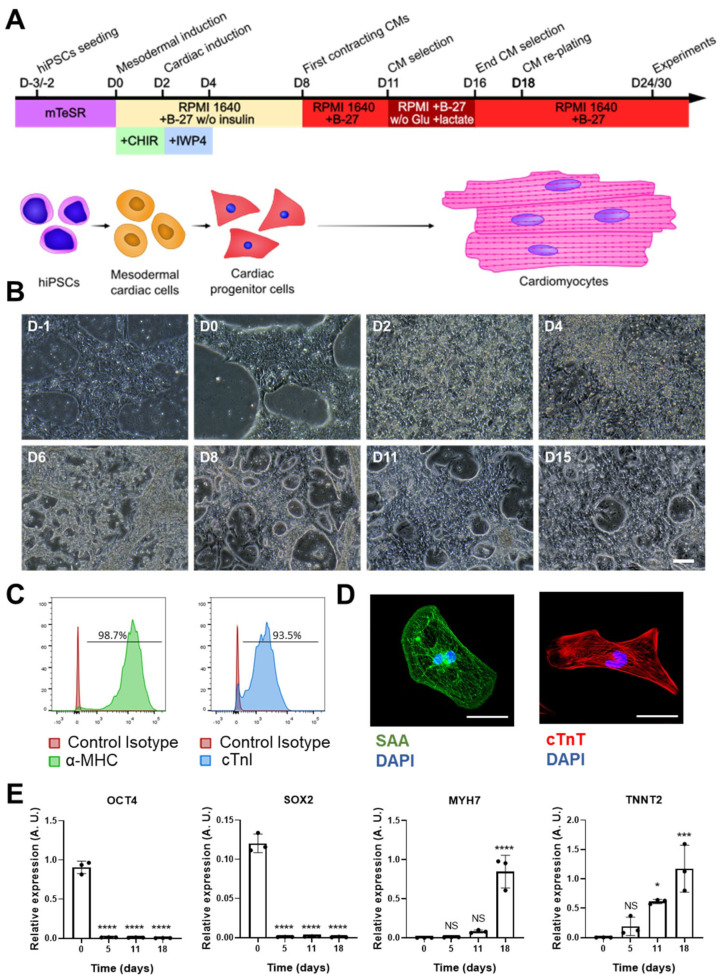
hiPSC-CM differentiation and characterization. (**A**) Scheme of the cardiac differentiation program. (**B**) Images of the cardiac differentiation of representative hiPSC-CMs: D-1 and D0 images show undifferentiated hiPSCs; the characteristic morphology of hiPSC-CMs can be observed from D8 onwards (scale bar = 100 μm). (**C**) Representative histograms of cardiac marker staining (myosin heavy chain (MHC) and cardiac troponin (cTnI)) of iPSC-CMs at day 25 of differentiation. (**D**) Representative images of hiPSC-CMs stained with antibodies against sarcomeric α-actinin and cardiac troponin T (scale bar = 50 µm). (**E**) Gene expression of pluripotency (*OCT4* and *SOX2*) and cardiac (*MYH7* and *TNNT2*) markers at different time points of cardiac differentiation. Data are represented as mean ± standard deviation. * *p* < 0.05, *** *p*< 0.001, **** *p* < 0.0001, NS = non-significant differences.

**Figure 2 antioxidants-12-01378-f002:**
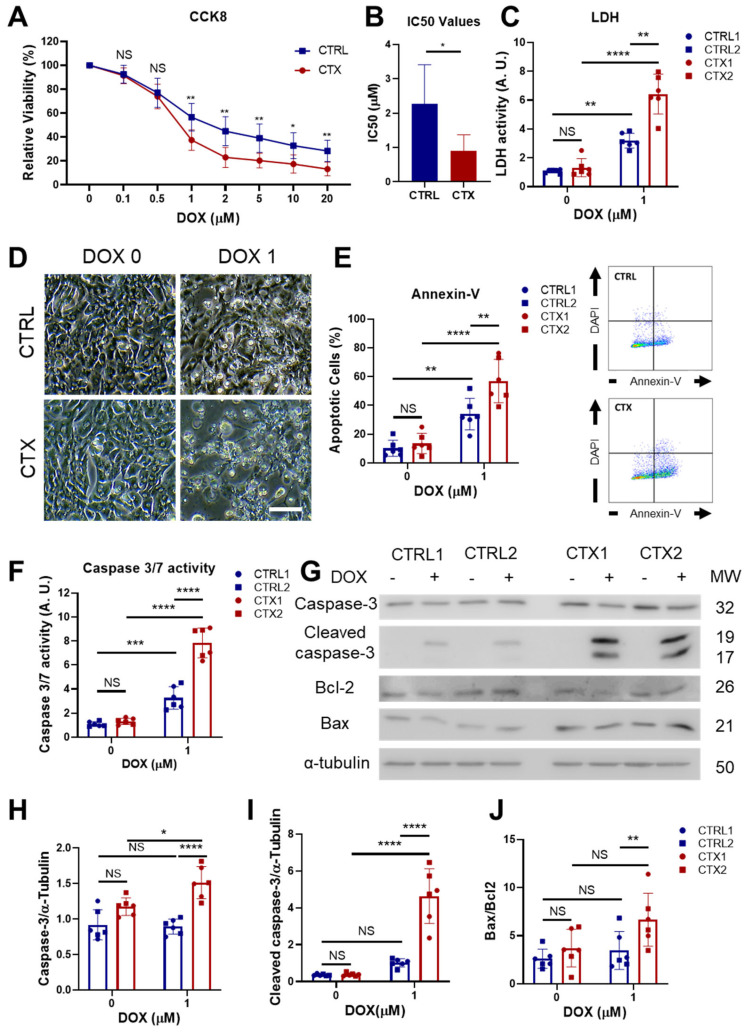
Effect of DOX on hiPSC-CM viability and apoptosis. (**A**) Viability curves of hiPSC-CMs after DOX exposure. (**B**) Half-maximal inhibitory concentration (IC_50_) of DOX in in CTRL and CTX hiPSC-CMs. (**C**) LDH activity measured in the culture medium before and after DOX treatment. (**D**) Representative phase-contrast microscopy images of CTRL and CTX cardiomyocytes before and after DOX treatment (scale bar = 100 µm). (**E**) Percentage of apoptotic cells in CTRL and CTX hiPSC-CMs before and after DOX treatment. Representative images of flow cytometry are shown on the right. (**F**) Caspase 3/7 activity measured in cell extracts of CTX and CTRL hiPSC-CMs before and after DOX treatment. (**G**) Representative Western blots of caspase-3, cleaved caspase-3, Bcl-2, Bax and α-tubulin of CTRL and CTX hiPSC-CMs treated or not with DOX. (**H**) Relative levels of caspase-3 were quantified by densitometry using α-tubulin as a loading control. (**I**) Relative levels of cleaved caspase-3 were quantified by densitometry using α-tubulin as a loading control. (**J**) Bax/Bcl-2 ratio quantified by densitometry. Data are represented as mean ± standard deviation. * *p* < 0.05, ** *p* < 0.01, *** *p* < 0.001 **** *p* < 0.0001, NS = non-significant differences.

**Figure 3 antioxidants-12-01378-f003:**
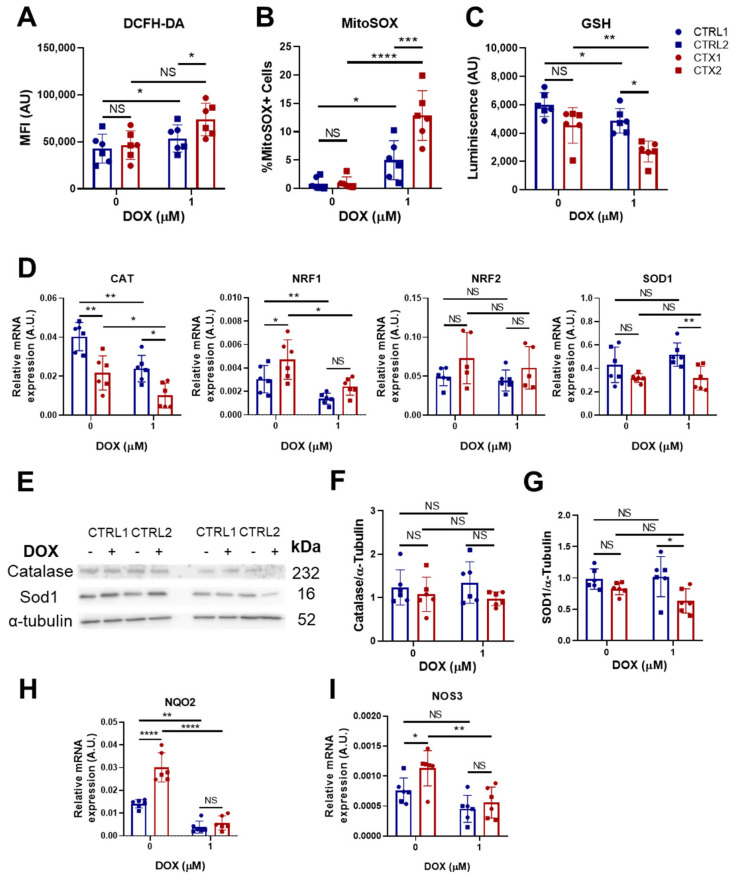
Effect of DOX on oxidative stress in hiPSC-CMs. (**A**) Quantification of total ROS in CTRL and CTX hiPSC-CMs treated or not with DOX. Cells were stained with DCFH-DA and analyzed by flow cytometry. (**B**) Quantification of mitochondrial superoxide in CTRL and CTX hiPSC-CMs treated or not with DOX. Cells were stained with MitoSOX^®^ and analyzed by flow cytometry. (**C**) Levels of GSH in CTRL and CTX hiPSC-CMs treated or not with DOX. (**D**) Expression levels of genes related to ROS detoxification in CTRL and CTX hiPSC-CMs treated or not with DOX. *GAPDH* expression was used as an endogenous control. (**E**) Representative Western blots of catalase, SOD1 and α-tubulin protein expression in CTRL and CTX hiPSC-CMs treated or not with DOX. (**F**) Relative levels of catalase were quantified by densitometry using α-tubulin as a loading control. (**G**) Relative levels of SOD1 quantified by densitometry using α-tubulin as a loading control. Expression levels of *NQO2* (**H**) and *NOS3* (**I**) related to detoxification and antioxidation processes, respectively, in CTRL and CTX hiPSC-CMs treated or not with DOX. *GAPDH* expression was used as an endogenous control. Data are represented as mean ± standard deviation. * *p* < 0.05, ** *p* < 0.01, *** *p* < 0.001, **** *p* < 0.0001, NS = not significant.

**Figure 4 antioxidants-12-01378-f004:**
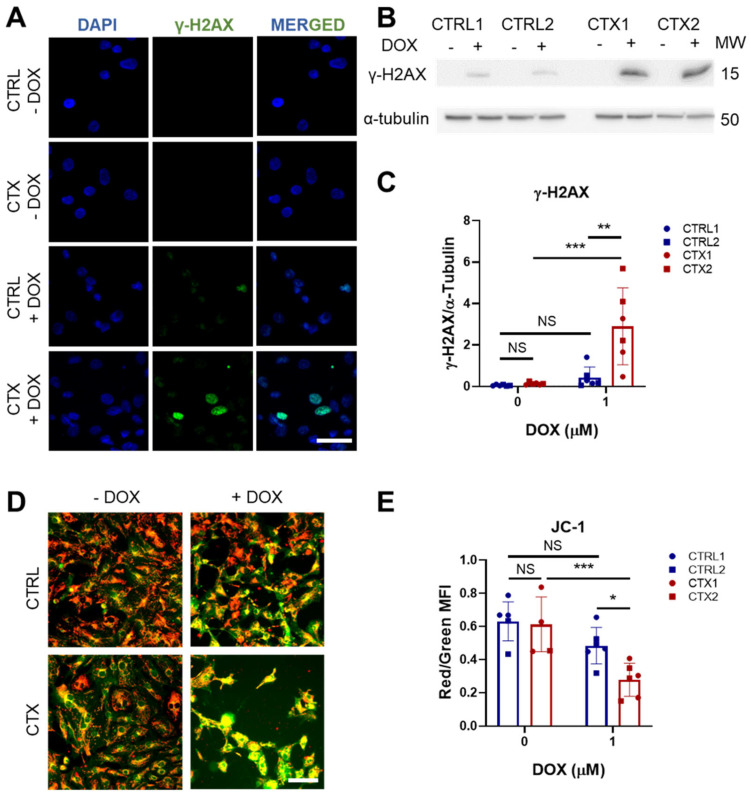
Effect of DOX on genotoxic stress and mitochondrial damage in hiPSC-CMs. (**A**) Representative immunofluorescence images of p-histone γ-H2AX staining (scale bar = 50 µm). Negative staining against the p-histone γ-H2AX in both CTRL and CTX hiPSC-CMs was observed before DOX treatment. (**B**) Representative Western blots of p-histone γ-H2AX expression hiPSC-CMs treated or not with DOX. (**C**) Quantification analysis using α-tubulin as a loading control. (**D**) Representative images of JC-1 staining in hiPSC-CMs treated or not with DOX. Merge of green and red channels is shown (scale bar = 100 µm). (**E**) Ratio between red and green fluorescence intensity quantified with ImageJ. Data are represented as mean ± standard deviation. * *p* < 0.05, ** *p* < 0.01, *** *p* < 0.001, NS = non-significant differences.

**Figure 5 antioxidants-12-01378-f005:**
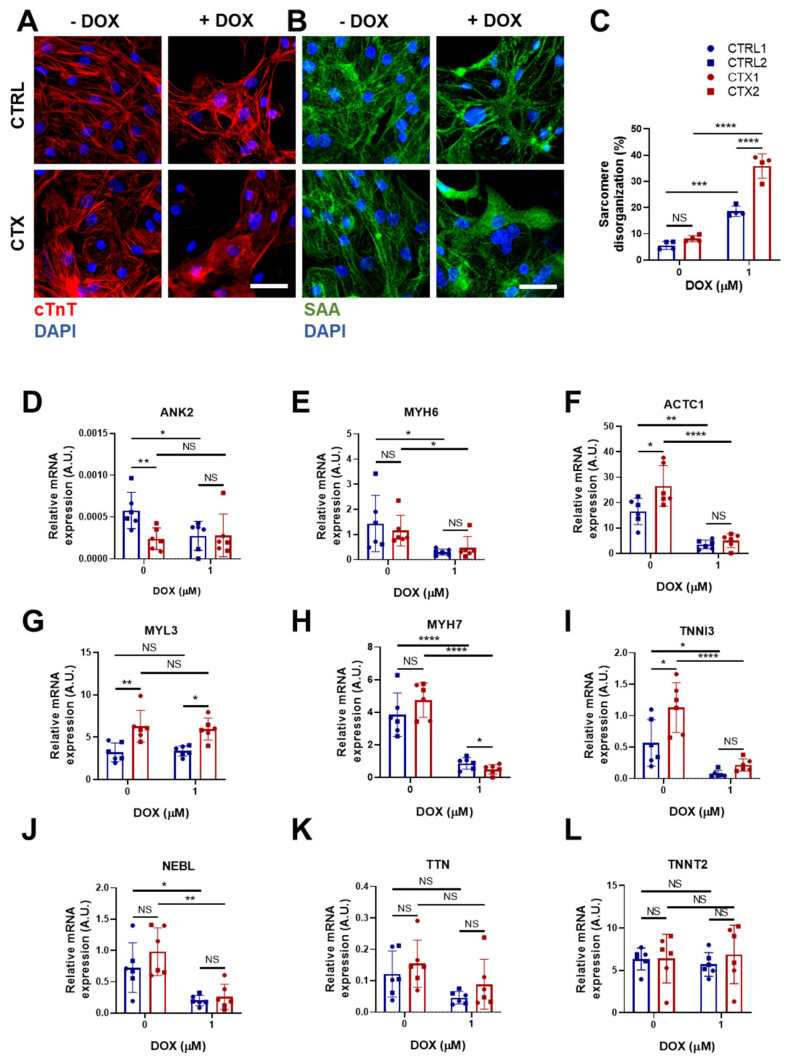
Effects of DOX on sarcomere organization and the expression of sarcomere genes in hiPSC-CMs. Representative immunostaining images of (**A**) cTnT and (**B**) sarcomeric α-actinin in hiPSC-CMs before and after DOX treatment (scale bar = 50 µm). (**C**) Percentage of sarcomere disorganization in hiPSC-CMs with and without DOX exposure, quantified using the images of sarcomeric α-actinin. (**D**–**L**) Relative expression of *ANK2, ACTC1*, *MYL3*, *TNNI3 MYH6*, *MYH7*, *NEBL*, *TNNT2* and *TTN*. We used *GAPDH* as an endogenous control. Data are represented as mean ± standard deviation. * *p* < 0.05, ** *p* < 0.01, *** *p* < 0.001, **** *p* < 0.0001, NS = non-significant differences.

**Figure 6 antioxidants-12-01378-f006:**
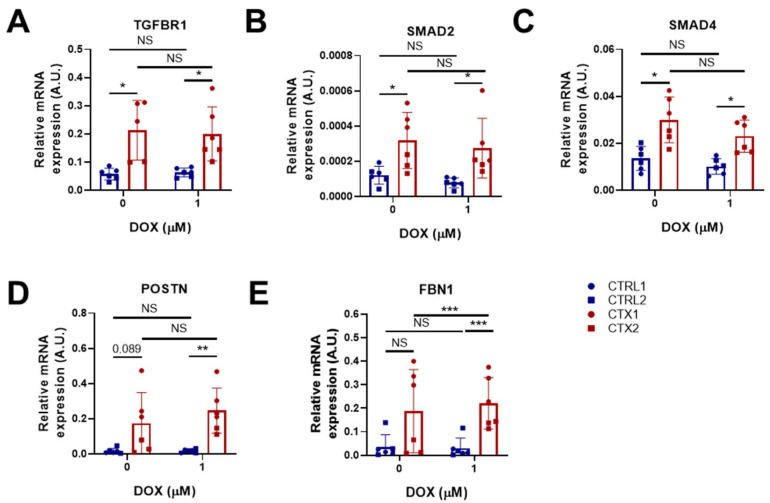
Effect of DOX on the expression of pro-fibrotic genes in hiPSC-CMs. (**A**–**E**) Relative expression of *TGFBR1, SMAD2, SMAD4*, *POSTN* and *FBN1*. We used *GAPDH* as an endogenous control. Data are represented as mean ± standard deviation. * *p* < 0.05, ** *p* < 0.01, *** *p* < 0.001, NS = non-significant differences.

**Figure 7 antioxidants-12-01378-f007:**
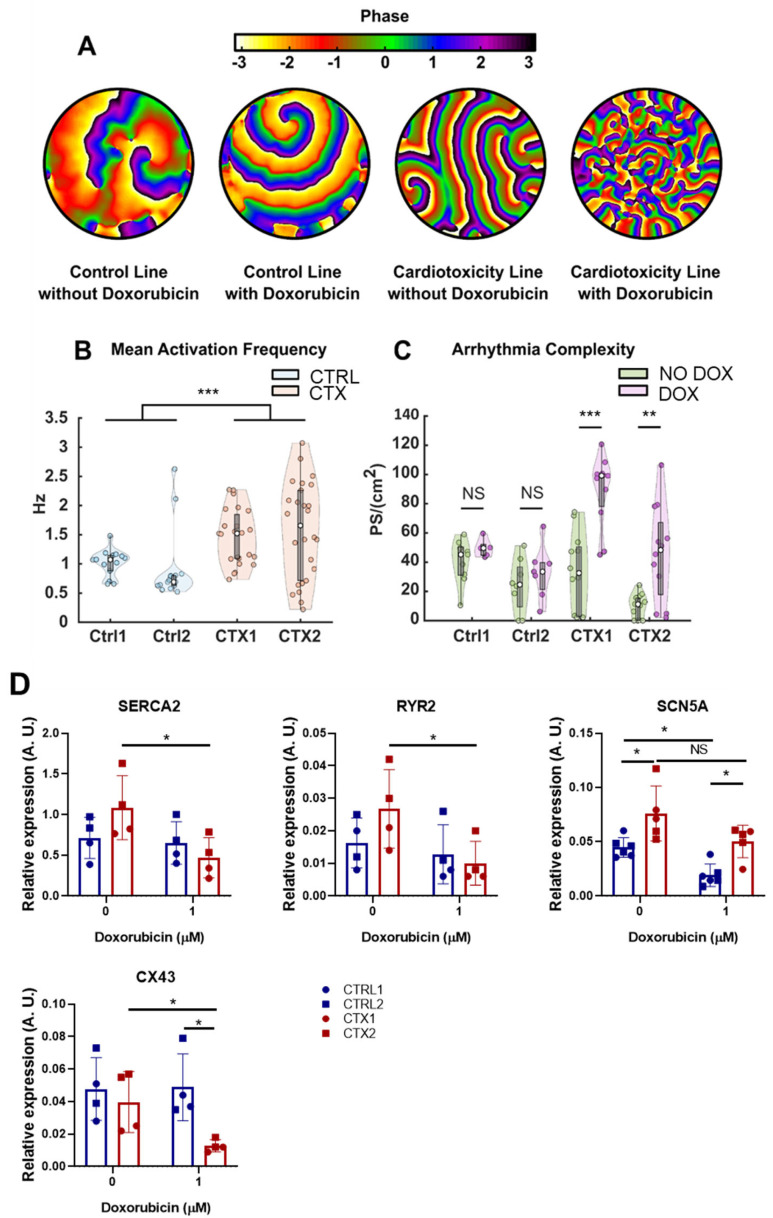
Effect of DOX on arrythmia complexity in hiPSC-CMs. (**A**) Example propagation patterns recorded in hiPSC-CMs under basal and DOX conditions. All samples are fibrillating, but less complex patterns are present in control lines. (**B**) Mean activation frequency obtained in hiPSC-CMs. (**C**) PS density values computed for all cell lines, with or without DOX, plotted separately. (**D**) Relative expression of *SERCA2*, *RYR2*, *SCN5A* and *Cx43*. GAPDH was used as an endogenous control. Data are represented as mean ± standard deviation. * *p* < 0.05, ** *p* < 0.01, *** *p* < 0.001, NS = non-significant differences.

**Figure 8 antioxidants-12-01378-f008:**
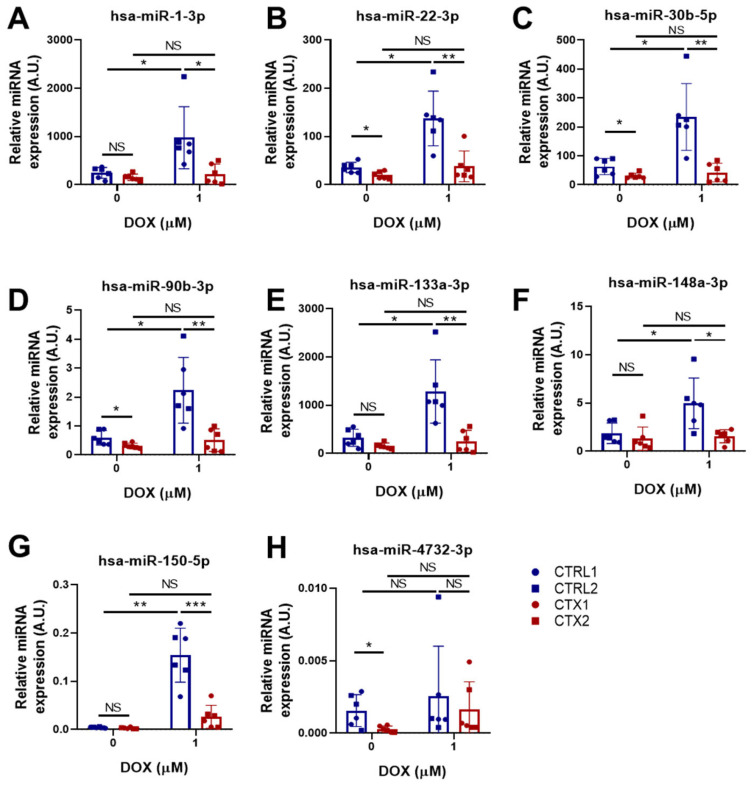
Effect of DOX on the expression levels of cardioprotective miRNAs in hiPSC-CMs. (**A**–**H**) Relative expression of hsa-miR-1-3p, hsa-miR-22-3p, hsa-miR-30b-5p, hsa-miR-90b-3p, hsa-miR-133a-3p, hsa-miR-148a-3p, hsa-miR-150-5p and hsa-miR-4732-3p. We used the snRNA U6 as an endogenous control. Data are represented as mean ± standard deviation. * *p* < 0.05, ** *p* < 0.01, *** *p* < 0.001, NS = non-significant differences.

**Table 1 antioxidants-12-01378-t001:** Patient demographic data.

	CTX	Sex	Type of Tumor	Cumulative AC Dose	Age at Diagnosis	Age at iPSC Generation	Tissue
CTRL 1	No	M	None	-	-	9	Foreskin
CTRL 2	No	M	None	-	-	2.5	Scrotum
CTX 1	Yes	M	Embryonal rhabdomyosarcoma	>150 mg/m^2^	3.9	4	PBMCs
CTX 2	Yes	M	AML	>250 mg/m^2^	14	14	PBMCs

CTX: cardiotoxicity; AML: acute myeloid leukemia; PBMCs: peripheral blood mononuclear cells; AC: Anthracycline; iPSCs: induced pluripotent stem cells.

**Table 2 antioxidants-12-01378-t002:** Number of optical mapping videos recorded and analyzed in cells under basal and DOX conditions.

	Control		Cardiotoxicity		
Cell Lines	CRTL 1	CTRL 2	CTX1	CTX2	Total
No DOX	9 (3)	8 (4)	12 (4)	14 (5)	42
DOX	5 (3)	7 (3)	11 (3)	12 (5)	36
Total	14	15	23	26	78

## Data Availability

Raw data of this work are available upon request by email to corresponding author.

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
