# Peer review of "Modeling Cardiotoxicity in Pediatric Oncology Patients Using Patient-Specific iPSC-Derived Cardiomyocytes Reveals Downregulation of Cardioprotective microRNAs"

_antioxidants, 2023, doi:10.3390/antiox12071378_

Round 1
Reviewer 1 Report
The manuscript by Reinal and coworkers delves into the mechanisms of doxorubicin-induced cardiotoxicty by using hiPSC-CMs derived from pediatric patients with clinical signs of cardiotoxicity. The authors compare the outcome of doxo exposure in these cells and those obtained from hiPSC lines from healthy patients. The manuscript covers several areas related to doxo-induced cardiotoxicity, such as oxidative stress, sarcomeric structure or arrhythmogenicity. I would like to congratulate the authors for putting together this interesting and sound piece of science.
As a whole, I do not have major comments. There are some minor issues/questions that I would like to see addressed:
1. Methods: the methodology for sarcomeric dissarray quantification is not present (or else I have missed it). Authors should include it in the appropriate section.
2. The authors mention throughout the manuscript that the CMs they derive are of a ventricular phenotype. However, this is not supported by the CM characterization data (action potential and differential gene/protein expression specially). It is true that the authors quote a protoclo (ref#26) for ventricular CM generation, but this is not proven by the authors nor (from what I have been able to gather) their publication track as referenced in the manuscript. In consequence, I suggest deleting the "ventricular" tag.
3. Sometimes the authors explicit the results for each cell line separatedly, but sometimes they do not (see for example Fig 2A and B). Can the authors explain this choice?
4. Table 1 needs formatting.
5. In my opinion, it would be best if Table 2 showed the number of independent experiments/differentiations analysed.
English language is used appropriatedly throughout the manuscript, although some typos remain. No doubt they will be edited during proofreading.
Reviewer 2 Report
The manuscript by Reinal et al. describe the generation of two distinct IPSC lines of pediatric oncology patients with enhanced susceptibility to cardiotoxicity upon anthracyclines treatment. The authors nicely demonstrate that CM derived from those IPCs display increased susceptibility to apoptosis, DNA damage, oxidative stress, mitochondrial and sarcomeric damage as well as arrhythmias upon DOX treatment. Additionally they also reported impaired microRNA expression.
Overall, the study is very well designed and highly relevant on the field, yet there are several points that should be considered.
While there are convincing evidences presented by the authors that apoptosis, DNA and mitochondrial damage is observed in these iPSC-derived cardiomyocytes, evidences on ROS impairment and sarcomeric disarray are more spurious. ROS markers do not always display the same trend and/or behavior upon treatment and the authors distinctly used such changes to justify their interests. Similarly, changes in sarcomeric proteins are not at all convincing on the images provided, probably because of the immature mature of the cells, as nicely stated in the conclusions/limitations subheading. Thus I would suggest to the authors to slow down their claims on these biological aspects.
Secondly, I am surprise that the authors nicely demonstrate a distinct arrhythmogenic pattern in the experimental IPSC-CM upon DOX treatment but they failed to investigate any plausible molecular mechanisms, i.e. are key cardiac action potential ion channels altered? it could be very interesting if the authors could further elaborate on this front.
Thirdly the same also is applicable for the microRNA analyses. Distinct microRNA expression is illustrated but no intend to dissect their molecular mechanisms is provided.
Reviewer 3 Report
In their manuscript, the authors conducted an interesting study on influencing the greatest reduction during the use of anthracyclines (on the example of doxorubicin). The study was conducted in-house, the results clear, well-discussed.
I actually only have one remark. As DOX is a commonly used drug in both children and adults, please:
- extending the introduction and discussing the mechanisms of cardiotoxicity, taking into account the course of the disease in different age groups,
- occurrence of DOX-disease interaction in these groups,
- as the study concerns acute toxicity, please comment (maybe a comparison in the table) in the introduction of the mechanisms and course of acute and commonly occurring delayed toxicity.
- please pay attention to the youngest class of mediators nitric oxide and carbon monoxide and the guanylate cyclase/cGMP signaling pathway in the development of this toxicity.
Round 2
Reviewer 3 Report
The authors made significant modifications to the manuscript. It may now be considered for publication